# Prospective Validation of Pentraxin-3 as a Novel Serum Biomarker to Predict the Risk of Prostate Cancer in Patients Scheduled for Prostate Biopsy

**DOI:** 10.3390/cancers13071611

**Published:** 2021-03-31

**Authors:** Ugo Giovanni Falagario, Gian Maria Busetto, Giuseppe Stefano Netti, Francesca Sanguedolce, Oscar Selvaggio, Barbara Infante, Elena Ranieri, Giovanni Stallone, Giuseppe Carrieri, Luigi Cormio

**Affiliations:** 1Department of Urology and Organ Transplantation, University of Foggia, 71122 Foggia, Italy; ugofalagario@gmail.com (U.G.F.); info@oscarselvaggio.it (O.S.); giuseppe.carrieri@unifg.it (G.C.); luigi.cormio@unifg.it (L.C.); 2Clinical Pathology Unit, Department of Medical and Surgical Sciences, University of Foggia, 71122 Foggia, Italy; giuseppestefano.netti@unifg.it (G.S.N.); elena.ranieri@unifg.it (E.R.); 3Department of Pathology, University of Foggia, 71122 Foggia, Italy; francesca.sanguedolce@unifg.it; 4Nephrology Dialysis and Transplantation Unit, University of Foggia, 71122 Foggia, Italy; barbara.infante@gmail.com (B.I.); giovanni.stallone@unifg.it (G.S.); 5Department of Urology, Bonomo Teaching Hospital, 70031 Andria, Italy

**Keywords:** prostate cancer, prostate biopsy, biomarkers, decision curve analysis, pentraxin 3

## Abstract

**Simple Summary:**

PTX3 could be a potential biomarker for PCa. To predict anyPCa and csPCa biomarker PTX3 was evaluated before PBx. Among the 455 eligible patients, PCa was detected in 49% and csPCa in 25%. PTX3 outperformed other variables in predicting both anyPCa and csPCa. Serum PTX3 levels might be of clinical utility in predicting prostate biopsy results.

**Abstract:**

Purpose: To test and internally validate serum Pentraxin-3 (PTX3) levels as a potential PCa biomarker to predict prostate biopsy (PBx) results. Materials and Methods: Serum PSA and serum PTX3 were prospectively assessed in patients scheduled for PBx at our Institution due to increased serum PSA levels or abnormal digital rectal examination. Uni- and multivariable logistic regression analysis, area under the receiver operating characteristic curve (AUC), and decision curve analysis (DCA), were used to test the accuracy of serum PTX3 in predicting anyPCa and clinically significant PCa (csPCa) defined as Gleason Grade (GG) ≥ 2. Results: Among the 455 eligible patients, PCa was detected in 49% and csPCa in 25%. During univariate analysis, PTX3 outperformed other variables in predicting both anyPCa and csPCa. The addition of PTX3 to multivariable models based on standard clinical variables, significantly increased each model’s predictive accuracy for anyPCa (AUC from 0.73 to 0.82; *p* < 0.001) and csPCa (AUC from 0.79 to 0.83; *p* < 0.001). At DCA, PTX3, and PTX3, density showed higher net benefit than PSA and PSA density and increased the net benefit of multivariable models in deciding when to perform PBx. Conclusions: Serum PTX3 levels might be of clinical utility in predicting prostate biopsy results. Should our findings be confirmed, this novel reflex test could be used to reduce the number and burden of unnecessary prostate biopsies.

## 1. Introduction

The gold standard for the diagnosis of prostate cancer (PCa) is prostate biopsy (PBx), but the diagnostic yield of this procedure remains low. In current clinical practice, the cancer detection rate (CDR) of a first extended PBx prompted by an elevated serum prostate-specific antigen (PSA) level or an abnormal digital rectal examination (DRE), is in the range of 40% and drops down to approximately 25% in the setting of screening programs, i.e., in patients with serum PSA between 2.5 and 10 ng/mL [1,2]. Moreover, even though it is a routine outpatient procedure, PBx is not free from complications that might even be severe [3].

Efforts to improve the diagnostic yield of PBx are oriented towards the use of risk calculators based on several clinical parameters, imaging techniques, particularly prostate multiparametric magnetic resonance imaging (mp-MRI), and serum or urine-based biomarkers. Up to 2019, EAU guidelines [4] provided a strong recommendation to offer asymptomatic men with normal DRE and serum PSA level < 10 ng/mL further assessment tools, such as a risk calculator, a prostate mp-MRI, and an additional serum or urine biomarker. The 2020 EAU Guidelines [5] conversely provide, in such clinical situations, a strong recommendation for risk calculators and imaging, but a weak recommendation for additional biomarkers, probably due to the fact that none of those discovered in the recent years proved to perform so well to enter clinical practice.

Pentraxin-3 (PTX3) belongs to the pentraxin superfamily, which are essential components of the humoral arm of the innate immune system and play a pivotal role in vascular biology [6,7]. Like short pentraxins, PTX3 facilitates dysregulation of mitogenic signaling pathways, sustains cellular proliferation, angiogenesis, insensitivity to apoptosis, cancer cell invasion and migration, and tumor escape from immunosurveillance [8,9]. Unlike short pentraxins, PTX3 is produced by a variety of cell types at the site of inflammation [6,10].

Considering that chronic inflammation is found in as much as 80% of PBxs [11] and that PTX3 seems to be involved in inflammatory-related carcinogenesis [12], we elected to test the potential correlation between PTX3 and PCa. In our first study, we demonstrated that PTX3 was overexpressed by prostate epithelial cells and extensively colocalized with PSA expressed by these cells. PTX3 also colocalized with myeloperoxidase-positive infiltrating cells, although they did not represent a major source of PTX3 within the prostate tissue, whereas CD4+ or CD20+ infiltrating cells did not significantly express the long pentraxin. Additionally, PTX3 was found to be overexpressed by prostate epithelial cells in patients who were diagnosed with PCa at second biopsy [13]. In a recent study, we tested PTX3 protein expression and complement activation in 80 patients with biopsy-proven benign prostate, who underwent second PBx 12 to 36 months later. Patients diagnosed with PCa at second biopsy showed overexpression of PTX3/C1q deposits, increased expression of C3a and C5a receptors, inactivation of the terminal complement complex C5b-9, and a significant increase of complement inhibitor CD59. Although we cannot be sure whether such association leads to a definite cause–effect relationship, data support the hypothesis that PTX3 might play a significant pathogenic role in PCa development by modulating complement activation [14].

Our first study [13] also showed that PTX3 serum levels were significantly higher in patients with PCa (6.1 ± 4.9 vs. 2.4 ± 1.1 ng/mL; *p* = 0.001), thus, pointing at PTX3 as a potential serum PCa biomarker. The present study therefore aimed to test and internally validate the accuracy of serum PTX3 in predicting PBx results.

## 2. Materials and Methods

The present study was carried out onto consecutive patients scheduled for transrectal ultrasound (TRUS) guided PBx at our institution, between January 2016 and December 2017, due to increased serum PSA levels (≥3 ng/mL) or abnormal DRE. The study was approved by the local Ethical Committee (Decision n. 152/CE/2014 of 3 September 2014; Ethical Committee at the University Hospital “Ospedali Riuniti”, Foggia, Italy) and was carried out in agreement with the provisions of the Helsinki Declaration held in 1995. All patients provided written informed consent to take part and data were entered into our prospectively maintained Institutional Review Board-approved PBx Database.

Before any prostate manipulation, serum samples were drawn for PSA and PTX3 assessment, and frozen at −80 °C for subsequently creating assays in duplicate using a commercially available ELISA Kit (PTX3 ab214570), according to the manufacturer’s instructions (R & D Systems, Minneapolis, MN, USA).

After local non-infiltrative anaesthesia [15,16], TRUS was used to determine prostate and transition zone volume and to guide transrectal prostate sampling, according to our systematic 18-core biopsy scheme [17]. PSA density was calculated as total PSA (ng/mL) divided by prostate volume. Similarly, we computed PTX3 density (PTX3/prostate volume).

Patients with serum PSA > 20 ng/mL were excluded due to a high risk (>75%) of harboring PCa. Men with potentially confounding factors such as treatment with 5 alfa-reductase inhibitors (5-ARIs), previous invasive treatment for benign prostatic hyperplasia, and dwelling urethral catheters, were also excluded.

All specimens were evaluated by a senior uropathologist, blind to the PTX3 data, according to contemporary diagnostic criteria for high-grade prostatic intraepithelial neoplasia (HGPIN), atypical small acinar proliferation (ASAP) of prostate, and PCa [18,19].

### Statistical Analysis

Descriptive statistics was performed for the overall population and according to biopsy results. Continuous variables were reported as median and interquartile range and was compared by the Mann–Whitney *U*-test, whereas categorical variables were reported as rates and tested by the Fisher’s exact test or the chi-square test, as appropriate.

The primary study endpoint was to test the accuracy of serum PTX3, used alone or in addition to other clinical variables, in predicting anyPCa and clinically significant PCa (csPCa), defined as Gleason Grade (GG) ≥ 2. To do so, we used univariate and multivariate logistic regression analysis, the area under the receiver operating characteristic curve (AUC), and decision curve analysis (DCA) that finally estimated the net clinical benefit weighing the relative harm of a missed cancer by the benefit of avoiding an unnecessary biopsy.

The secondary study endpoint was to attempt a correlation of the PSA and PTX3 findings, particularly in the most common challenging scenario, namely patients with PSA in the grey-area (3–10 ng/mL).

Statistical analyses were performed using Stata 14 (StataCorp LP, College Station, TX, USA), following journal recommendation [20]. All tests were 2-sided with a significance level set at *p* < 0.05.

## 3. Results

Among the 576 patients scheduled for PBx, 455 patients met the inclusion criteria; their clinical characteristics are summarized in Table 1. Overall, 329 patients were biopsy-naïve (74%) and 283 patients had a negative DRE and underwent biopsy for an elevated PSA.

PCa was detected in 49% of patients and csPCa in 25%. Patients with csPCa but not those with GG1 cancer had significantly higher serum PSA levels than those without PCa. Conversely, patients with both GG1 and csPCa had significantly higher serum PTX3 levels than those without PCa (all *p*-values ≤ 0.001).

Univariate analysis pointed out that all clinical variables, including age, biopsy history, DRE, prostate volume, PSA, and PTX3 were significant predictors of anyPCa and csPCa, but PTX3 outperformed all other variables in predicting both anyPCa and csPCa (Table 2).

Similarly, PTX3 density outperformed PSA density (Table 2). Multivariate analysis confirmed that PTX3 was a significant predictor of anyPCa and csPCa, and pointed out that the addition of PTX3 to the model, based on the standard clinical variables significantly increased the model predictive accuracy for anyPCa (AUC from 0.73 to 0.82; *p* < 0.001) and csPCa (AUC from 0.79 to 0.83; *p* < 0.001) (Table 3).

Moreover, DCA pointed out that PTX3 and PTX3 density showed a higher net benefit than PSA and PSA density, and increased the net benefit of multivariate models in deciding when to perform PBx (Figure 1).

To project findings into clinical practice, we matched pathology results with PSA and PTX3 findings. Figure 2 and Table 4 show the probability of anyPCa (A) and csPCa (B), according to serum PSA and PTX3 levels.

The PSA “grey-zone” (3–10 ng/mL) and the PTX3 “grey-zone” (2–6 ng/mL) had similar overall detection rates for GG1 PCa (25.1% vs. 24.6%, respectively) and csPCa (22.4% vs. 22.5%, respectively). However, while in the PTX3 “grey-zone” a PSA 3–10 or above 10 ng/mL did not impact on the detection of anyPCa and csPCa, in the PSA “grey-zone” the PTX3 categories had a significant impact on the detection rates of both anyPCa and csPCa (Table 4). Findings were even more evident in patients with negative DRE, whereby the biomarker values play a key role.

## 4. Discussion

The potential role of PTX3 in PCa biology is receiving increasing attention. We already contributed to available knowledge by showing that PCa patients display tissue overexpression of PTX3/C1q deposits and of C3a and C5a receptors, inactivation of the terminal complement complex C5b-9, and a significant increase of complement inhibitor CD59, suggesting at least a strong association if not a definite cause–effect relationship between PTX3 tissue overexpression and PCa development, through modulation of complement activation [14]. We also showed [13] that PTX3 serum levels were significantly higher in patients with PCa, suggesting that, like PSA, there is an increased PTX3 release in the blood of patients with PCa.

The present prospective study confirmed our initial findings of serum PTX3 significantly outperforming serum PSA in predicting PBx results. Moreover, the addition of serum PTX3 to statistical models incorporating PSA and standard clinical variables, significantly improved the model diagnostic accuracy, thus being of potential value in reducing the number and harms of unnecessary PBxs.

Since non-malignant conditions such as prostatic inflammations and benign prostate hyperplasia might increase PSA levels, there is a great quest for an efficient second test, the so-called reflex test, which is able to increase the CDR of csPCa in patients with a PSA within the grey zone (3–10 ng/mL), while reducing the number of negative biopsy results and overdiagnosis of indolent PCa. Using PTX3 as a reflex test in patients with a normal DRE and PSA in the grey zone would result in 17% biopsy reduction (39/224), missing 9% of GG1 PCa (5/57) and no csPCa. Additionally, in our study, the overall CDR of GG1 tumors was 25.1%; ranging from 15.9% in those with low PTX3 levels to 37.1% in those with high PTX3 levels. In the same group, the overall CDR of csPCa was 22.4%, ranging from 3.2% in those with low PTX3 levels to 45.2% in those with highPTX3 levels. The wider discrepancy in CDR of csPCa (3.2% vs. 45.2%) compared to GG1 tumors (15.9% vs. 37.1%) further point at the relevance of PTX3 in detecting not only anyPCa but also csPCa.

Results of such additional tests might be used alone or included in multivariate models that take into account multiple clinical and laboratory variables.

Human Kallikreins other than PSA, such as free PSA, intact PSA and hK2, are the most used reflex tests for this purpose and are included in the currently available blood-based test for prostate cancer screening. 4Kscore^®^ Test (OPKO Health, Miami, FL, USA) was developed by Vickers et al. in 2008, using a large cohort of men from the Gothenburg arm of the European Randomized PCa Screening Study (ERSPC), and is based on a logistic model that considers four forms of kallikrein (total PSA, free PSA, intact PSA, and hK2), in order to accurately predict the presence of PCa in men with a PSA 3.0 ng/mL or higher (AUC 0.84) [21]. The validity of the model was confirmed in a separate ERSPC cohort (Rotterdam arm) [22]. The use of the 4K test could avoid 513 biopsies per 1000, missing only 54 of 177 low-grade tumors and 12 of 100 high-grade tumors [22].

Similarly, the Prostate Health Index (PHI) adds the level of an isoform of the pro-PSA [p2PSA], itself correlated to PCa, to the free/total PSA ratio. Three prospective multicenter studies showed that both the PHI and the 4K test had diagnostic performances higher than the f/t PSA test alone in men, with a PSA between 2–10 ng/mL [23,24,25].

The Stockholm-3 (STHLM3) model included clinical variables (age, first-degree family history of prostate cancer, and a previous biopsy), blood biomarkers (total PSA, free PSA, ratio of free/total PSA, hK2, MIC1, and MSMB), genetic markers (a genetic score based on 254 single-nucleotide polymorphisms [SNPs] and an explicit variable for the HOXB13 SNP), and prostate examination (DRE, and prostate volume) [26]. The STHLM3 study prospectively demonstrated a 32% reduction in biopsies using the STHLM3 model compared to PSA ≥ 3 ng/mL for biopsy referral, without any loss in sensitivity, by also detecting GG ≥ 2 cancers for the PSA range 1–3 ng/mL and similar findings were achieved in the external validation of a multi-ethnic cohort [26,27].

While the above-mentioned biomarkers involved several complex laboratory tests on genomic profiling for detecting SNPs, our study showed the potential utility of a novel biomarker that could be easily dosed in the blood of patients at risk of PCa, with low additional costs. Our findings need, of course, to be externally validated and externally compared with the other available reflex tests.

Our study might have potential limitations. First, it is a single center study that obviously needs, as mentioned above, external validation. Second, it explored the role of PTX3 only as a reflex test as it was carried out in patients who were already scheduled for PBx based on increased PSA levels or abnormal DRE. Third, it was carried out in the pre mp-MRI era. Therefore, the role of this potential reflex test needs to be evaluated in view of the current mp-MRI diagnostic pathway [28]. However, given the high negative predictive value of PTX3 alone and in combination with PSA, we expect to demonstrate its usefulness as a triage test, as shown for the 4k test [29] and the STHLM3 model [30], in patients with PSA in the grey zone, to decide whether to perform mp-MRI and, eventually, PBx.

## 5. Conclusions

The present study demonstrates the clinical utility of PTX3 serum levels as a PCa biomarker. PTX3 serum levels outperformed PSA alone in predicting results of PBx. The addition of PTX3 to available statistical models incorporating PSA and standard clinical variables also significantly increased their diagnostic accuracy. These findings suggest that this novel reflex test has the potential to reduce the number and burden of unnecessary prostate biopsies.

## Figures and Tables

**Figure 1 cancers-13-01611-f001:**
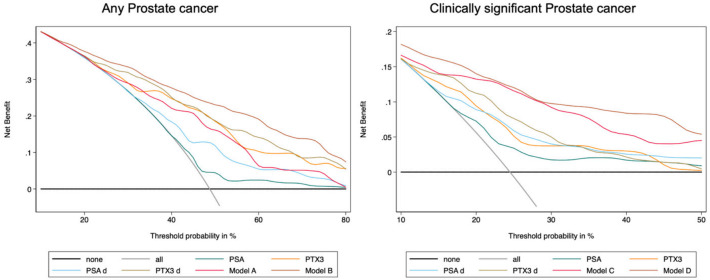
Decision curve analysis (DCA) comparing clinical utility of PTX3, PSA, PSA density, and PTX3 density for detecting any prostate cancer and csPCa. The DCA simulates two scenarios—one in which all patients would receive biopsy (all) and one in which none undergoes biopsy (none). Clinically useful models lie above these scenarios. PTX3 alone and models including PTX3 (Model B and Model D) showed a higher net benefit at each threshold probability and thus outperformed PSA alone and model including only standard variables in determining the need for a prostate biopsy.

**Figure 2 cancers-13-01611-f002:**
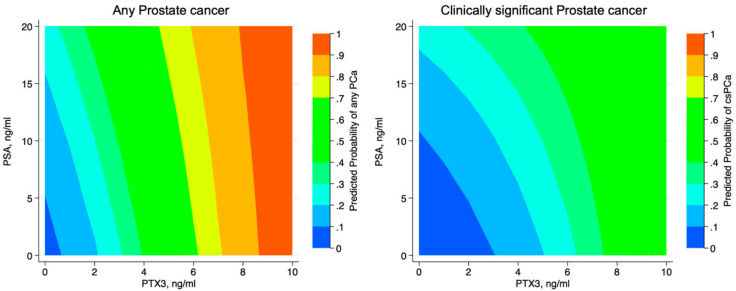
Contour plot showing the predicted probability of anyPCa and csPCa in patients undergoing prostate biopsy according to PSA and PTX3. For PSA values below 10 ng/mL, PTX3 is able to better discriminate presence of anyPCa and csPCa.

**Table 1 cancers-13-01611-t001:** Clinical characteristics of the overall population stratified according to biopsy results.

Patients	Overall (*N* = 445)	No Cancer (*N* = 228)	GG1 (*N* = 108)	GG ≥ 2 (*N* = 109)	*p*-Value
Age (yr)	66.1 (60.0, 71.9)	64.7 (58.2, 70.7)	65.2 (60.0, 69.7)	69.4 (64.5, 73.4)	**<0.0001**
Biopsy history					
*Naive*	329 (73.9%)	155 (68.0%)	77 (71.3%)	97 (89.0%)	**0.0002**
*Previous neg.*	116 (26.1%)	73 (32.0%)	31 (28.7%)	12 (11.0%)	
DRE					
*Negative*	283 (63.6%)	157 (68.9%)	74 (68.5%)	52 (47.7%)	**0.0004**
*Suspicious*	162 (36.4%)	71 (31.1%)	34 (31.5%)	57 (52.3%)	
Prostate Volume (ml)	54.0 (40.0, 70.0)	60.0 (46.2, 81.5)	52.0 (36.0, 66.0)	41.0 (35.0, 55.0)	**<0.0001**
PSA (ng/mL)	6.6 (5.0, 9.3)	6.4 (4.8, 8.4)	6.4 (4.8, 9.4)	7.6 (5.8, 11.3)	**0.001**
PTX3 (ng/mL)	3.4 (2.4, 5.2)	2.7 (1.9, 3.6)	4.3 (3.1, 6.2)	4.3 (3.3, 6.9)	**<0.0001**
PSA density	0.12 (0.09, 0.18)	0.11 (0.08, 0.14)	0.12 (0.09, 0.19)	0.16 (0.11, 0.26)	**<0.0001**
PTX3 density	0.06 (0.04, 0.11)	0.04 (0.03, 0.07)	0.08 (0.05, 0.15)	0.10 (0.06, 0.17)	**<0.0001**

GG—Gleason grade group; and DRE—digital rectal examination; In bold significant *p*-values.

**Table 2 cancers-13-01611-t002:** Univariate logistic regression predicting anyPCa and csPCa.

Covariates	Univariate Analysis Predicting anyPCa				Univariate Analysis Predicting csPCa			
	OR	95% CI	*p* > |z|	AUC	OR	95% CI	*p* > |z|	AUC
Age (yr)	1.03	1.01, 1.05	**0.009**	0.58	1.05	1.03, 1.08	**<0.001**	0.64
Biopsy history				0.56				0.60
*Naive*	Ref.				Ref.			
*Previous neg.*	0.52	0.34, 0.81	**0.004**		0.28	0.15, 0.52	**<0.001**	
DRE				0.55				0.61
*Negative*	Ref.				Ref.			
*Suspicious*	1.60	1.08, 2.36	**0.018**		2.41	1.55, 3.75	**<0.001**	
Prostate Volume (mL)	0.97	0.97, 0.98	**<0.001**	0.67	0.97	0.96, 0.98	**<0.001**	0.67
PSA (ng/mL)	1.05	1.02, 1.09	**0.003**	0.57	1.07	1.03, 1.10	**<0.001**	0.61
PTX3 (ng/mL)	1.64	1.46, 1.85	**<0.001**	0.76	1.35	1.22, 1.50	**<0.001**	0.69
PSA density	1.79	1.42, 2.25	**<0.001**	0.67	1.57	1.31, 1.87	**<0.001**	0.70
PTX3 density	7.83	4.81, 12.74	**<0.001**	0.79	3.25	2.26, 4.66	**<0.001**	0.74

DRE—digital rectal examination; In bold significant *p*-values.

**Table 3 cancers-13-01611-t003:** Multivariate logistic regression predicting anyPCa and csPCa.

Covariates	Any Prostate cancer (GG ≥1)						Clinically Significant Prostate Cancer (GG ≥ 2)					
	Multivariate Model A AUC = 0.73			Multivariate Model B AUC = 0.82			Multivariate Model C AUC = 0.79	Multivariate Model D AUC = 0.83				
	OR	95% CI	*p* > |z|	OR	95% CI	*p* > |z|	OR	95% CI	*p* > |z|	OR	95% CI	*p* > |z|
Age	1.03	1.00, 1.05	**0.015**	1.02	1.00, 1.05	**0.047**	1.05	1.02, 1.08	**<0.001**	1.05	1.02, 1.08	**0.001**
Biopsy History												
*Naive*	Ref.			Ref.			Ref.			Ref.		
*Previous Neg.*	0.60	0.37, 0.96	**0.035**	0.65	0.39, 1.10	**0.109**	0.30	0.15, 0.61	**0.001**	0.32	0.16, 0.65	**0.002**
DRE												
*Negative*	Ref.			Ref.			Ref.			Ref.		
*Positive*	1.49	0.98, 2.28	**0.063**	1.73	1.08, 2.75	**0.022**	2.33	1.42, 3.81	**0.001**	2.48	1.49, 4.15	**0.001**
PSA	1.09	1.04, 1.14	**<0.001**	1.08	1.03, 1.14	**0.001**	1.10	1.05, 1.15	**<0.001**	1.09	1.04, 1.15	**<0.001**
Prostate vol.	0.97	0.96, 0.98	**<0.001**	0.97	0.96, 0.98	**<0.001**	0.97	0.96, 0.98	**<0.001**	0.97	0.96, 0.99	**<0.001**
PTX3				1.63	1.43, 1.85	**<0.001**				1.31	1.17, 1.47	**<0.001**

GG—Gleason grade group; DRE—digital rectal examination; In bold significant *p*-values.

**Table 4 cancers-13-01611-t004:** Number of cancers by prostate specific antigen (PSA) and Pentraxin3 (PTX3) levels.

All Patients	PTX3 < 2 ng/mL	PTX3 2–6 ng/mL	PTX3 > 6 ng/mL	Overall
PSA < 3 ng/mL	(*N* = 7)	(*N* = 11)	(*N* = 2)	*N* = 20
*No cancer*	7 (100.0%)	7 (63.6%)	0 (0.0%)	14 (70.0%)
*GG* = 1	0 (0.0%)	2 (18.2%)	1 (50.0%)	3 (15.0%)
*GG* ≥ 2	0 (0.0%)	2 (18.2%)	1 (50.0%)	3 (15.0%)
PSA 3–10 ng/mL	(*N* = 63)	(*N* = 214)	(*N* = 62)	*N* = 339
*No cancer*	51 (81.0%)	116 (54.2%)	11 (17.7%)	178 (52.5%)
*GG* = 1	10 (15.9%)	52 (24.3%)	23 (37.1%)	85 (25.1%)
*GG* ≥ 2	2 (3.2%)	46 (21.5%)	28 (45.2%)	76 (22.4%)
PSA > 10 ng/mL	(*N* = 15)	(*N* = 51)	(*N* = 20)	*N* = 86
*No cancer*	10 (66.7%)	23 (45.1%)	3 (15.0%)	36 (41.9%)
*GG* = 1	1 (6.7%)	14 (27.5%)	5 (25.0%)	20 (23.3%)
*GG* ≥ 2	4 (26.7%)	14 (27.5%)	12 (60.0%)	30 (34.9%)
Overall	(*N* = 85)	(*N* = 276)	(*N* = 84)	
*No cancer*	68 (85%)	146 (52.9%)	14 (16.7%)	
*GG* = 1	11 (12.9%)	68 (24.6%)	29 (34.5%)	
*GG* ≥ 2	6 (7%)	62 (22.5%)	41 (48.8%)	
**Negative DRE**	**PTX3 < 2 ng/mL**	**PTX3 2–6 ng/mL**	**PTX3 > 6 ng/mL**	**Overall**
PSA < 3 ng/mL	(*N* = 4)	(*N* = 7)	(*N* = 0)	*N* = 11
*No cancer*	4 (100.0%)	5 (71.4%)	0 (0.0%)	9 (81.8%)
*GG* = 1	0 (0.0%)	2 (28.6%)	0 (0.0%)	2 (18.2%)
*GG* ≥ 2	0 (0.0%)	0 (0.0%)	0 (0.0%)	0 (0.0%)
PSA 3–10 ng/mL	(*N* = 39)	(*N* = 144)	(*N* = 41)	*N* = 224
*No cancer*	34 (87.2%)	86 (59.7%)	10 (24.4%)	130 (58.0%)
*GG* = 1	5 (12.8%)	34 (23.6%)	18 (43.9%)	57 (25.4%)
*GG* ≥ 2	0 (0.0%)	24 (16.7%)	13 (31.7%)	37 (16.5%)
PSA > 10 ng/mL	(*N* = 8)	(*N* = 28)	(*N* = 12)	*N* = 48
*No cancer*	5 (62.5%)	13 (46.4%)	0 (0.0%)	18 (37.5%)
*GG* = 1	1 (12.5%)	10 (35.7%)	4 (33.3%)	15 (31.2%)
*GG* ≥ 2	2 (25.0%)	5 (17.9%)	8 (66.7%)	15 (31.2%)
Overall	(*N* = 51)	(*N* = 179)	(*N* = 53)	
*No cancer*	43 (84.3%)	104 (58.1%)	10 (18.9%)	
*GG* = 1	6 (11.8%)	46 (25.7%)	22 (41.5%)	
*GG* ≥ 2	2 (3.9%)	29 (16.2%)	21 (39.6%)	

GG—Gleason grade group; DRE—digital rectal examination.

## Data Availability

The data presented in this study are available on request from the authors.

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
