# Peer review of "Prospective Validation of Pentraxin-3 as a Novel Serum Biomarker to Predict the Risk of Prostate Cancer in Patients Scheduled for Prostate Biopsy"

_cancers, 2021, doi:10.3390/cancers13071611_

Round 1
Reviewer 1 Report
The authors provide statically aided evidence for the role of PTX3Â as a suitable marker for prediction of prostrate cancer. While the role of serum PTX3 is evaluated, the authors may want to comment on site specific tumor PTX3 or immune cell PTX3 levels in the discussion for future studies or other groups.Â
Author Response
Sir,
Thank you very much for having reviewed our manuscript and having given us the possibility to further improve it on the basis of the precious comments provided by the Reviewers. We sincerely appreciated their work and made our best to address their comments. Changes to the manuscript have been tracked as requested and have been enclosed below each reply to Reviewer comments. Â Â
We hope this new version will be considered suitable for publication in your Journal and look forward to hearing from you in this respect at your soonest convenience.
As for REVIEWER # 1 comments:
1.0 The authors provide statically aided evidence for the role of PTX3 as a suitable marker for prediction of prostate cancer. While the role of serum PTX3 is evaluated, the authors may want to comment on site specific tumor PTX3 or immune cell PTX3 levels in the discussion for future studies or other groups.
REPLY 1.0: We thank the reviewer for the positive comments on our paper. Additionally, we really appreciate the opportunity to clarify the important aspect of the localization and expression of PTX3. Indeed, in our previous study, using double immunofluorescence confocal microscopy, we demonstrated that PTX3 extensively colocalized with PSA expressed by prostate epithelial cells. PTX3 also colocalized with Myeloperoxidase+ infiltrating cells, although they did not represent a major source of PTX3 within the prostate tissue. On the other hand, CD4+ or CD20+ infiltrating cells did not significantly express the long pentraxin. Please find below the revised paragraph of the introduction:
Â
“In our first study we demonstrated that PTX3 was overexpressed by prostate epithelial cells and extensively colocalized with PSA expressed by these cells. PTX3 also colocalized with Myeloperoxidase-positive infiltrating cells, although they did not represent a major source of PTX3 within the prostate tissue, whereas CD4+ or CD20+ infiltrating cells did not significantly express the long pentraxin.
Reviewer 2 Report
Unnecessary prostate biopsy and repeat PSA measurement in non-PCa patients with PSA level of under 10~20 ng/ml are a problem that needs to be resolved. The authors prospectively evaluated serum Pentraxin-3 (PTX3) levels as a potential PCa biomarker to predict prostate biopsy (PBx) results. At univariable logistic regression, PTX3 outperformed other variables in predicting both anyPCa (OR 1.79) and csPCa (OR 1.57). At multivariable logistic regression, PTX3 with standard clinical variables significantly increased each model predictive accuracy for anyPCa (AUC from 0.73 to 0.82; p<0.001) and csPCa (AUC from 0.79 to 0.83; p<0.001). At decision curve analysis (DCA), PTX3 alone seemed to show higher net benefit than PSA alone in csPCa even in the lower threshold probability (15~20%). Model D, PTX3 with standard clinical variables, showed the highest net benefit in both anyPCa and csPCa. The overall quality of the writing is excellent. As a reviewer, I point out some problems about your article as follow.
Â
Problems
- Your study included the patients with previous negative biopsy. The detection rates of anyPCa and csPCa in patients with previous negative biopsy were lower than those with naĂŻve biopsy. Did the inclusion of patients with previous negative biopsy affect the results of your study?
- I have a difficulty to understand Table 4. The authors stratified patients according to Gleason grade groups (GG) such as any GG (anyPCa) or GG>=2 (csPCa) in the analyses for Table 2, Table 3 and Figure 1. However, in Table 4, the authors stratified patients with GG1 and anyGG (GG>=1). I don’t understand why the authors stratified patients in the different way in Table 4. In addition, what does the color mean in Table 4? Why the number of patients with GG1 is higher than those with >=GG1?
- Regarding X-axis and Y-axis, the range of minimum to maximum in Figure 1 with anyPCa should be the same as Figure 1 with csPCa.
- In Figure 1, I agree with you in which models including PTX3 (Model B and Model D) showed a higher net benefit at each threshold probability. How about the PTX3 alone and PTX3 density alone? The authors said that “PTX3 and PTX3 density showed an higher net benefit than PSA and PSA density” in patients with both anyPCa and csPCa. However, PTX3 alone and PTX density alone seemed to have no net benefit at less than 20% threshold probability in anyPCa when compared with PSA alone. Some clinical variables such as higher age and co-morbidities affect the threshold probability to perform prostate biopsy. How much threshold probability do you think suitable for performing prostate biopsy in patients with PSA level less than 20ng/ml in the setting of your study? Is the threshold probability in above 20% enough to show the net benefit for PTX3 in anyPCa?
- One of the purposes in your study is to reduce the number of unnecessary prostate biopsy. At the same time, we should avoid detecting non-csPCa in daily practice. While PTX3 measurement likely helps to predict anyPCa, it seems like to help the detection for GG1 patients in the results of Table 4. GG1 patients with lower PSA level are candidate for active surveillance. Does PTX3 measurement help to detect non-csPCa or help to exclude non-csPCa? Does higher levels of PTX3 related to both the detection for csPCa and non-csPCa? The group of anyPCa included a lot of GG1 patients (n=108). How do the readers of this journal understand the results of anyPCa in Figure 1, Figure 2, and Table 4 in these points? The authors should describe about these points in Discussion part.
- How do you calculate PSA density and PTX3 one? You should mention about the methods in your article.
- Overall, the authors should discuss more about the results of their study rather than most used reflex tests such as 4Kscore and PHI in Discussion part.
Author Response
Sir,
Thank you very much for having reviewed our manuscript and having given us the possibility to further improve it on the basis of the precious comments provided by the Reviewers. We sincerely appreciated their work and made our best to address their comments. Changes to the manuscript have been tracked as requested and have been enclosed below each reply to Reviewer comments. Â Â
We hope this new version will be considered suitable for publication in your Journal and look forward to hearing from you in this respect at your soonest convenience.
As for REVIEWER # 2 comments:
2.0 Unnecessary prostate biopsy and repeat PSA measurement in non-PCa patients with PSA level of under 10~20 ng/ml are a problem that needs to be resolved. The authors prospectively evaluated serum Pentraxin-3 (PTX3) levels as a potential PCa biomarker to predict prostate biopsy (PBx) results. At univariable logistic regression, PTX3 outperformed other variables in predicting both anyPCa (OR 1.79) and csPCa (OR 1.57). At multivariable logistic regression, PTX3 with standard clinical variables significantly increased each model predictive accuracy for anyPCa (AUC from 0.73 to 0.82; p<0.001) and csPCa (AUC from 0.79 to 0.83; p<0.001). At decision curve analysis (DCA), PTX3 alone seemed to show higher net benefit than PSA alone in csPCa even in the lower threshold probability (15~20%). Model D, PTX3 with standard clinical variables, showed the highest net benefit in both anyPCa and csPCa. The overall quality of the writing is excellent. As a reviewer, I point out some problems about your article as follow.
REPLY 2.0: We thank the reviewer for the precise summary of our work and for the positive feedback. We tried to address each comment carefully and we believe the new version of the manuscript improved significantly. Thank you. Â
2.1 Your study included the patients with previous negative biopsy. The detection rates of anyPCa and csPCa in patients with previous negative biopsy were lower than those with naĂŻve biopsy. Did the inclusion of patients with previous negative biopsy affect the results of your study?Â
REPLY 2.1: Thank you for this comment. We agree with the reviewer that cancer detection rates after a first negative biopsy are lower and, in this scenario, additional biomarkers to predict presence of cancer are even more important. To assess if the inclusion of patients with previous negative biopsy affected the results of our study, we performed a sub analysis stratifying patients based on the biopsy setting (Table 4). Additionally, biopsy history was included in the multivariable analysis (Table 3) and PTX3 kept its significance. Thank you.Â
2.2 I have a difficulty to understand Table 4. The authors stratified patients according to Gleason grade groups (GG) such as any GG (anyPCa) or GG>=2 (csPCa) in the analyses for Table 2, Table 3 and Figure 1. However, in Table 4, the authors stratified patients with GG1 and anyGG (GG>=1). I don’t understand why the authors stratified patients in the different way in Table 4. In addition, what does the color mean in Table 4? Why the number of patients with GG1 is higher than those with >=GG1?
REPLY 2.2: We thank the reviewer for pointing this out and we apologize for the mistake. GG≥1 in table 4 actually refer to GG≥2 namely clinically significant prostate cancer. Indeed, patients were stratified in No cancer, GG 1 and GG≥2 across all the paper. Thank you for pointing this out, we modified table 4 accordingly.
2.3 Regarding X-axis and Y-axis, the range of minimum to maximum in Figure 1 with anyPCa should be the same as Figure 1 with csPCa.
REPLY 2.3: We thank the reviewer for her/his comment. We agree with the reviewer regarding the utility of having similar ranges for x- and y- axes. However, as you can imagine, the prevalence of AnyPCa is higher than the prevalence of csPCa thus the net benefit of a model predicting AnyPCa is definitely higher than the benefit of a model predicting csPCa. Different ranges are needed to graphically present our findings without reducing images dimensions. To improve the readability of the image we kindly ask reviewer and editor to keep Figure 1 as it is.
2.4 In Figure 1, I agree with you in which models including PTX3 (Model B and Model D) showed a higher net benefit at each threshold probability. How about the PTX3 alone and PTX3 density alone? The authors said that “PTX3 and PTX3 density showed an higher net benefit than PSA and PSA density” in patients with both anyPCa and csPCa. However, PTX3 alone and PTX density alone seemed to have no net benefit at less than 20% threshold probability in anyPCa when compared with PSA alone. Some clinical variables such as higher age and co-morbidities affect the threshold probability to perform prostate biopsy. How much threshold probability do you think suitable for performing prostate biopsy in patients with PSA level less than 20ng/ml in the setting of your study? Is the threshold probability in above 20% enough to show the net benefit for PTX3 in anyPCa?
REPLY 2.4: Thank you for this comment. Figure 1 represent a decision curve analysis which is, as you correctly pointed out, a statistical method to compute the net benefit of applying a diagnostic test to the decision to perform a procedure at different test cut-offs of probability. However rather than deciding what is the range of cutoffs of PTX3 or PTX3 density to perform a biopsy in patients at risk of prostate cancer, we used decision curve analysis to compare PTX3 net benefit to PSA net benefit which represent the current gold standard. As you can see PTX3 and PTX3 density outperformed PSA and PSA density at each cutoff. Thank you for your comment.
2.5 One of the purposes in your study is to reduce the number of unnecessary prostate biopsy. At the same time, we should avoid detecting non-csPCa in daily practice. While PTX3 measurement likely helps to predict anyPCa, it seems like to help the detection for GG1 patients in the results of Table 4. GG1 patients with lower PSA level are candidate for active surveillance. Does PTX3 measurement help to detect non-csPCa or help to exclude non-csPCa? Does higher levels of PTX3 related to both the detection for csPCa and non-csPCa? The group of anyPCa included a lot of GG1 patients (n=108). How do the readers of this journal understand the results of anyPCa in Figure 1, Figure 2, and Table 4 in these points? The authors should describe about these points in Discussion part.
 REPLY 2.5: Thank you for the opportunity to clarify this aspect.
The utility of PTX3 measurement is clearly seen in Table 4. Specifically, in patients with PSA 3-10 ng/ml the overall CDR of GG1 tumors is 25.1%; adding PTX3 levels, it ranges from 15.9% in those with low PTX3 levels to 37.1% in those with high PTX3 levels. In the same group, the overall CDR of csPCa is 22.4% ranging from 3.2% in those with low PTX3 levels to 45.2% in those with highPTX3 levels. The wider discrepancy in CDR of csPCa (3.2% vs 45.2%) compared to GG1 tumors (15.9% vs 37.1%) further point at the relevance of PTX3 in detecting not only PCa but the cs ones!
Finally, since PTX3 is involved in the early stages of PCa development, we believe it is going to be useful as a prognostic marker for GG1 patients who are going to progress on AS. Â A study evaluating outcomes of patients with GG1 and 2 PCa is currently ongoing. Please find below the comment we added in the discussion section. Thank you.
“Since non-malignant conditions such as prostatic inflammations and benign pros-tate hyperplasia may increases PSA levels, there is a great quest for an efficient second test, the so-called reflex test, able to increase CDR of csPCa in patients with a PSA with-in the grey zone (3-10 ng/ml) while reducing the number of negative biopsy results and overdiagnosis of indolent PCa. Using PTX3 as a reflex test in patients with a nor-mal DRE and PSA in the grey zone would result in 17% biopsy reduction (39/224) miss-ing 9% of GG1 PCa (5/57) and no csPCa. Additionally, in our study the overall CDR of GG1 tumors was 25.1%; ranging from 15.9% in those with low PTX3 levels to 37.1% in those with high PTX3 levels. In the same group, the overall CDR of csPCa was 22.4% ranging from 3.2% in those with low PTX3 levels to 45.2% in those with highPTX3 lev-els. The wider discrepancy in CDR of csPCa (3.2% vs 45.2%) compared to GG1 tumors (15.9% vs 37.1%) further point at the relevance of PTX3 in detecting not only any PCa but also csPCa.
Results of such additional test may be used alone or included in multivariable models that take into account multiple clinical and laboratory variables.”
2.6 How do you calculate PSA density and PTX3 one? You should mention about the methods in your article.
REPLY 2.6: We thank the reviewer for this comment. Please find below and in the methods section the added sentence:
“PSA density was calculated as total PSA (ng/ml) divided by prostate volume. Similarly, we computed PTX3 density (PTX3/prostate volume).”
2.7 Overall, the authors should discuss more about the results of their study rather than most used reflex tests such as 4Kscore and PHI in Discussion part.
REPLY 2.7: We thank the reviewer for her/his comment. As suggested, and thanks to your precious previous comments we updated the manuscript with additional discussion point on our findings.